# What helps and hinders reproducible research? Researchers' perspectives from a cross-disciplinary interview study

Magdalena Kozula[1]*, Nicholas J. DeVito[2], Patrick Onghena[1], Cinzia Colombo[3], Leonie A. Dudda[4], Paula Muñoz Teno[1], Veerle Van den Eynden[5¤]

**1** Faculty of Psychology and Educational Sciences, KU Leuven, Leuven, Flanders, Belgium, **2** Nuffield Department of Primary Care Health Sciences, University of Oxford, Oxford, United Kingdom, **3** "Research with citizens for prevention and care" Unit, Laboratory of Lifestyle Research, Department of Medical Epidemiology, Istituto di Ricerche Farmacologiche Mario Negri IRCCS, Milan, Italy, **4** Department of Otorhinolaryngology and Head & Neck Surgery, University Medical Center Utrecht, Utrecht, The Netherlands, **5** Research Coordination Office, KU Leuven, Leuven, Flanders, Belgium

¤ Current address: Directorate Research, Library and Internationalisation, Hasselt University, Hasselt, Flanders, Belgium
* magdalena.kozula@kuleuven.be

## Abstract

Debates and policy initiatives addressing research reproducibility have expanded considerably in recent years. Yet, many of these measures remain generic and risk overlooking the lived realities of research practice. This study aims to explore researchers' perspectives on the barriers, facilitators, and motivators that shape reproducible research across diverse fields and career stages, using qualitative methods. Semi-structured interviews were conducted with 60 researchers affiliated with universities and research institutions across the European Union and the United Kingdom. Participants were sampled to ensure diversity in discipline, career stage, gender, and geography. The interviews explored experiences with barriers and facilitators for reproducible research, and the data were analyzed using framework analysis with a hybrid inductive–deductive approach. Five interrelated themes described barriers and facilitators influencing reproducibility: navigating the research ecosystem (incentives and policies of institutions, journals, and funders); social and cultural dynamics as drivers and barriers (disciplinary norms, generational differences, competition, and collaboration); resourcing reproducibility (skills, infrastructure, guidelines and standards, time, funding, and awareness); inside the research process (field-specific constraints, methodological transparency, research material sharing, and external restrictions); and personal commitment to shared responsibility (reflective motivations, pragmatic drivers, and perceptions of accountability). Researchers described reproducibility as less of an individual choice but as a socially and institutionally mediated activity, dependent on enabling conditions such as supportive policies, adequate infrastructure, and equitable resource distribution. Reproducibility reform cannot rely solely on individual researcher commitment or one-size-fits-all

**Data availability statement:** The underlying data, consisting of 60 interview transcripts, has been published in the KU Leuven Research Data Repository: https://doi.org/10.48804/FMK8FK. Of the 60 transcripts, 58 are openly accessible. The remaining 2 are available upon request, in accordance with the participants' wishes, due to potentially sensitive participant information, as these participants shared details and opinions that can still be identified despite the pseudonymisation and that can influence their careers. Requests for access may be submitted via the repository or directed to the KU Leuven Research Data Management support service at rdm@kuleuven.be.

**Funding:** This work was supported by the project OSIRIS, funded by the European Union's Horizon Europe research and innovation program under grant agreements No. 101094725. The funders had no role in study design, data collection and analysis, decision to publish, or preparation of the manuscript.

**Competing interests:** The authors have declared that no competing interests exist.

policies. Effective interventions must account for disciplinary and methodological diversity, provide targeted resources and training, and realign incentive structures to reward transparency and rigour. These findings highlight reproducibility as a collective responsibility across the research ecosystem, requiring coordinated action by researchers, institutions, funders, and publishers. Promoting reproducible practices in this systemic, context-sensitive manner is essential for fostering a more credible, equitable, and sustainable scientific enterprise.

## Introduction

In recent years, concerns about the credibility of scientific research have brought the issue of reproducibility to the forefront of scholarly and policy debate. Reproducibility speaks to long-standing ideals about how science should operate – as a collective and critically self-correcting enterprise. Ensuring that findings can be independently reproduced not only strengthens the credibility of individual studies but also reinforces these institutional ideals, safeguarding the fairness, openness, and scrutiny on which scientific progress depends.

The salience of reproducibility has intensified amid high-profile failures to reproduce results in fields such as psychology and biomedical sciences [1,2]. These problems have opened up a space for reform, prompting a wave of policy-level and community-driven initiatives aimed at improving research quality and transparency [3]. Key among these are reforms associated with the open science (OS) movement, including initiatives that promote pre-registration, methodological transparency, sharing of study data and materials, and reporting standards [4,5]. While this ambition is widely shared, a recent scoping review on reproducibility interventions by Dudda and colleagues [6] investigated the evidence on OS initiatives and found that it remains limited. These reforms are increasingly seen as overly generic, offering solutions that do not sufficiently account for diverse domains in which research is designed, conducted, and evaluated [7,8].

Such concerns necessitate translating policy ambitions to the everyday conduct of research, addressing the factors that impact reproducibility at their root. This cannot be achieved without attention to how individual researchers across different contexts – disciplinary, institutional, geographical, etc. – perceive, interpret, and experience reproducibility, including its barriers, motivations, and enablers. Their central and active roles in day-to-day research make researchers essential to understanding how reproducibility is enacted, challenged, and adapted within the domain-specific research workflows they execute [9,10]. This consideration has been echoed in recent calls to account for epistemic diversity to ensure the effectiveness of reforms [11–13]. Many recommended practices and advice for their implementation have been shaped by a relatively narrow set of disciplines, like psychology and health sciences, where reproducibility issues first gained prominence – and born out in the focus of subsequent research clustering within these fields [6]. This disciplinary concentration, along with its predominantly Western orientation [11] and variations in

institutional cultures, governance, and resources, may limit the applicability of such practices across the broader research landscape. If implemented as one-size-fits-all solutions, they risk unintended consequences, such as exacerbating inequities within the research system [14,15].

A growing body of research has examined barriers to, and facilitators of, reproducibility-enhancing and OS practices, using quantitative, mixed, and qualitative designs. Surveys consistently show that many researchers support transparency and reproducibility in principle, yet uptake remains uneven and constrained by multiple mechanisms. First, feasibility constraints reflect broad resource gaps such as time, skills, access to guidance, and long-term stewardship support [16–23]. Second, incentive and governance constraints include publish-or-perish pressures [24–28], limited rewards and recognition for reproducibility-related labour [22,29], and novelty-oriented journal cultures [30]. Several studies also identified fear as an important barrier in OS practices, highlighting concerns about data misuse, competition, and being scooped [16,19,31,32]. Across studies, reported facilitators mirror these barriers, emphasising resourcing – mentorship and training, technical measures and tools [21,23,33] and academic recognition as crucial in decisions on engaging in, for example, data sharing practices [17,19,34,35].

Qualitative research both corroborates and extends these survey-based patterns. In a meta-synthesis of qualitative studies on data sharing and reuse, Perrier et al. [36] identify recurring constraints – time and resourcing, perceived risks and uncertainty, and incentive misalignment. Interview and focus group studies add explanatory value by showing how barriers and facilitators are produced and negotiated in specific research settings through the design of local systems, routines, and material arrangements that condition what is feasible in practice [37,38]. Qualitative studies also help identify context-sensitive leverage points for intervention design by clarifying which facilitators must be organisationally implemented (e.g., resource hubs, outreach channels, internal review processes, resource allocation; [37,39–42]). Finally, design-oriented qualitative work highlights requirements for sustainable support by showing how interventions must generate practical benefits for researchers and fit existing workflows to be adopted over time [43,44].

At the same time, the evidence base remains fragmented and uneven in coverage. Much empirical work focuses on a limited set of practices, most prominently data sharing and related OS behaviours [16–19,31,32,34,40–42,45]. Methodologically, a substantial part of studies concentrates on specific populations and settings, including discipline-specific communities [30–32,38,40,43,44] and early-career researchers [20,24]. Several studies are also bound to single institutions or single-country contexts (e.g., [18,19,40,41,43]). This can limit the ability to connect findings across research contexts and to examine how barriers and facilitators interact across the wider research ecosystem.

To address this gap, we draw on 60 interviews with researchers across disciplines, career stages, and institutional settings in Europe, to examine how reproducibility is experienced and navigated in everyday research practice. Our study set out to examine reproducibility in a broad policy-facing sense. Much of the existing empirical literature focuses on open scholarship in general or on specific proxy practices (such as data sharing), whereas studies explicitly centred on reproducibility tend to address barriers and facilitators within narrower, practice- or field-specific problems. This article presents the barriers that hinder reproducibility, the motivations that encourage researchers to adopt reproducible practices, and the drivers or enablers that can strengthen their implementation. It is embedded in the project OSIRIS: Open Science to Increase Reproducibility in Science, which aims to support the shift toward reproducibility and inform European research policy by understanding its drivers, testing evidence-based solutions, identifying stakeholder incentives, and integrating reproducibility into the research design.

By employing an applied qualitative approach [46,47], we aimed to provide a better understanding of reproducibility as a process from the standpoint of participants [48]. Rather than treating barriers as isolated factors, we examined collected data looking at how reproducibility is shaped across multiple levels – through external governance and incentives, social and disciplinary cultures, resourcing and infrastructure, and the practical realities of conducting research – thereby clarifying both shared patterns and context-specific leverage points for policy and intervention design. These insights can inform more context-sensitive and equitable reforms, contributing to a more inclusive, reflexive, and sustainable culture of reproducible research.

## Methods

### Study design

This was an explorative, qualitative study using semi-structured, one-to-one interviews on barriers and facilitators to reproducibility. The aim was to recruit a heterogeneous group of researchers with balanced representation of disciplines, career stages, gender, and demographic diversity across Europe. Given that our findings are intended to inform European Commission policies on research, we limited recruitment to researchers affiliated with research-performing institutions based in the European Union (EU) and the United Kingdom (UK). The study was reported using the Consolidated Criteria for Reporting Qualitative Research Checklist (S1 Appendix).

Two minor deviations from the preregistered protocol occurred [49]. First, for final-stage recruitment (last 7 interviewees), instead of contacting chosen departments within our sampled institutions as a secondary tool to identify researchers to invite to participate, we utilised consortium project contacts to address diversity gaps. Second, for the saturation assessment, while we applied the concepts of code and meaning saturation as planned, the evaluation was integrated iteratively into rolling analysis rather than through the fully structured tracking outlined in the protocol.

### Participants and recruitment

The target population was active researchers – defined as having a recent indexed publication and an affiliation with one of our target institutions – working at higher education institutions and research entities in EU member states or the UK that had received Horizon 2020 funding. Participants were recruited by applying stratified sampling with multistage purposeful random sampling [50] for a balanced sample across career stages, gender, research disciplines and geography. We initially selected 20 institutions using a stratified random approach based on European subregion and level of Horizon 2020 funding (top 10 recipients per region vs. other recipients in the region = remainders). Within institutions, researchers were identified via Web of Science by assessing the affiliations of first authors of recent publications and invited by personalised email explaining the study's aims.

Recruitment occurred between 17 November 2023 and 18 December 2024. It proceeded in stages, with sample composition monitored after each tranche to address imbalances. Detailed information about the recruitment strategy, including our primary sampling units, tools utilised, and applied quotas, is available in the supplementary material (S2 Appendix).

Our target was to interview 50–60 participants, with the final number dependent on reaching code and meaning saturation in the collected data as described by Hennink and colleagues [51]. Of 249 invited researchers, 60 agreed and were interviewed. The rest included non-responders and those who declined to participate in the study, primarily because of time constraints. We collected demographic information about researchers who accepted the interview invitation: research field (classified according to FORD Fields of Research and Development classification; [52]), career stage as defined in the European Commission [53], years in research and gender. Participant demographics are presented in Table 1.

Our general goal was to achieve a balanced sample in terms of gender, career stage, geography and discipline. As shown in Table 1, the sample is reasonably distributed across the six FORD disciplinary groups, with no single field dominating the dataset, although social sciences form the largest group. Accordingly, the findings should be interpreted as cross-disciplinary patterns emerging from a heterogeneous sample rather than as a basis for systematic between-discipline comparison.

### Data collection

We used an interview guide developed from the research questions [54,55], relevant theoretical frameworks [56], and pilot testing with four volunteer interviewees from our institutions (MK, NJD, VVdE, LD) that were not included in the final analysis. Open-ended questions invited participants to elaborate freely on their experiences [57,58]. Given the disciplinary diversity of participants, tailored probes were used to revisit and expand on relevant narrative elements [59,60]. Particular care was taken with the order and accessibility of language in the interview guide to support participant comprehension

**Table 1. Sample Demographics (N = 60).**

| Parameter | Definition | N | % |
|---|---|---|---|
| **Gender** | | | |
| **Female** | Identifies as a female researcher | 29 | 48 |
| **Male** | Identifies as a male researcher | 31 | 52 |
| **Career stage** | | | |
| **Early career** | Carry out research under supervision; Master's degree and PhD students | 14 | 23 |
| **Recognised** | PhD holders or equivalent who are not yet fully independent | 10 | 17 |
| **Established** | Researchers who have developed a level of independence | 21 | 35 |
| **Leading** | Researchers leading their research area or field | 15 | 25 |
| **Discipline** | | | |
| **Natural sciences** | Mathematics, computer and information, physical, chemical, earth and related environmental, biological and other natural sciences | 11 | 18 |
| **Engineering and technology** | Civil, electrical, electronic, information, mechanical, chemical, materials, medical, environmental engineering, environmental biotechnology, industrial biotechnology, nanotechnology, other engineering and technologies | 9 | 15 |
| **Medical and health sciences** | Basic medicine, clinical medicine, health sciences, health biotechnology, other medical sciences | 11 | 18 |
| **Agricultural sciences** | Agriculture, forestry and fisheries, animal and dairy science, veterinary science, agricultural biotechnology, other agricultural sciences | 5 | 8 |
| **Social sciences** | Psychology, economics and business, educational sciences, sociology, law, political science, social and economic geography, media and communications, other social sciences | 17 | 28 |
| **Humanities** | History and archaeology, languages and literature, philosophy, ethics and religion, arts, history of arts, performing arts, music, other humanities | 7 | 12 |
| **Region** | | | |
| **Northern Europe** | Denmark, Finland, Ireland, Sweden, UK | 16 | 27 |
| **Eastern Europe** | Hungary, Lithuania, Poland, Romania | 15 | 25 |
| **Southern Europe** | Croatia, Greece, Italy, Portugal, Spain | 15 | 25 |
| **Western Europe** | Belgium, France, Germany, Netherlands | 14 | 23 |
| **Institution** | | | |
| **Top 10** | Top 10 recipients of Horizon 2020 funding in the region by number of projects | 22 | 37 |
| **Remainder** | Other recipients of Horizon 2020 funding | 31 | 52 |

and expression [61]. The guide was developed to address a broader set of research objectives within the OSIRIS project work than those examined in the present article. It is available in the S3 Appendix.

One-to-one interviews, lasting approximately 60 minutes, took place between December 2023 and December 2024 and were held predominantly online via MS Teams platform. No repeat interviews were conducted. Three university-based researchers conducted the interviews: two females and one male (MK, MA; NJD, PhD; VVdE, PhD). All had prior experience with this mode of data collection. Prior to each interview, written informed consent was obtained online from participants. No relationship was established between the research team members and the interviewees prior to the interview, except for an introductory exchange of emails. Interviews were audio recorded, transcribed using MS Teams automatic transcription feature, followed by manual review and correction of transcripts by members of the interview team and PM. Interviewers did not take field notes. Conversations in foreign languages were translated into English by the interviewer for the purpose of analysis. Identifiable data were stored separately from transcripts, and all collected data was held securely on KU Leuven servers. Pseudonymised transcripts are deposited in the KU Leuven Research Data Repository. Prior to deposit, they were sent to participants for their review.

## Data analysis

We used framework analysis [62] with a hybrid inductive-deductive approach. Two researchers (MK, NJD) familiarised themselves with the first 20 interviews, developed an initial coding framework through open coding [63], and refined it iteratively. In the next step, four researchers (MK, NJD, VVdE, CC) independently coded a sub-sample of interviewers (10% of the dataset as of September 2024) with predefined 'meaning units' for analysis [64], to support inter-coder reliability (ICR) and inter-coder agreement (ICA). We operationalised ICR and ICA using a negotiated-agreement approach [64], following the process as featured in Hemmler et al. [65]. In line with this approach, we compared coding across coders and reviewed excerpts with partial agreement, especially those with 50% agreement or lower, for further calibration and possible codebook refinement. Rather than relying on a coefficient-based threshold, we assessed consistency through iterative calibration: coders explained their reasoning, discussed alternative interpretations, and refined code definitions until a shared understanding was reached. This process improved consistency in code application and strengthened the codebook by adding clarifications, examples, and non-examples, which was particularly useful for interpreting subtle meanings, where coders' different disciplinary and professional backgrounds helped identify code applications that others had missed.

Following calibration, remaining transcripts were single-coded in NVivo (MK), with 'Not Sure' codes and new emergent codes reviewed by the wider team (including PO). Charting and mapping steps aided by NVivo were used to synthesise and reorganise the coding framework to arrive at the final themes [66]. This process was guided by both deductive and abductive reasoning, which supported explanatory groupings of patterns and helped communicate the findings [67–69]. Participants were not asked to provide feedback on the findings in this study. The codebook developed for this study is available in the S4 Appendix.

In conceptualising reproducibility, we follow the European Commission's framing, where reproducibility is understood as a continuum "based on three main research processes: reproduction, replication, and re-use. All three processes rest on the availability of data and methods from the original study" [70]. Rather than treating these as separate categories, this approach views them as stages along a shared trajectory, all dependent on the availability of underlying data, methods, and documentation. This conceptualisation is grounded in the principle of transparency. Such a broad and process-oriented understanding enabled us to inclusively discuss barriers and facilitators with a varied sample. It also aligned with our use of semi-structured, open-ended interviews, which allowed participants to express diverse experiences shaped by the norms, constraints, and practices in their fields and settings.

## Research quality and reflexivity

We employed multiple strategies to ensure the credibility, transferability, dependability, and confirmability of this research, and applied active reflexivity, understood as aligning three levels of reflection: researchers' positionality, participants' perceptions of that positionality, and the assumptions we make about those perceptions and other factors built into the qualitative interviewing and coding process [71]. Because interview data are co-created between participant and interviewer, we remained attentive to how our backgrounds, experience, and assumptions could shape data generation and interpretation [72]. Interviews were conducted by multiple members of the research team and discussed in peer debriefings, reducing the likelihood that one interviewer's assumptions or interaction style would systematically shape the dataset. In addition, pilot testing of the interview guide, tailored probes, and real-time member checks during the conversations were used to support clarity and accurate capture of participants' intended meaning [72–74]. We recognise that reproducibility may be a normatively charged topic and therefore may invite socially desirable responses. To reduce this risk, interviews were conducted by university-based researchers who did not hold positions of authority over participants and were not acting as institutional authorities such as funders or managers. At the same time, our shared experience of conducting research helped us occupy an insider position, which supported

rapport with participants. Interviewees frequently provided critical and ambivalent accounts rather than uniformly favourable descriptions when discussing reproducibility and their practices. Reflexive awareness of the team's positionality, together with investigator triangulation during coding, was used to support transparency and rigour [75,76]. We drew on awareness of our positionality not only to scrutinise assumptions but also to deepen understanding of the topics discussed in the interviews, particularly where the team's varied disciplinary, professional, and geographical perspectives helped make sense of participants' accounts [77]. Detailed information about applied research quality strategies and positionality statements of the team members is available in the S5 Appendix.

## Results

Our analysis generated five main themes describing barriers and facilitators to reproducibility and open science. They are summarised in Table 2 and visually synthesised in Fig 1, which contrasts barriers and facilitators and situates them across the layers that can shape researchers' practice. We discuss the themes below with illustrative quotes from interviews. These may contain some minor linguistic mistakes, as the majority of our sample were not native English speakers, but these were not corrected unless they impacted readability.

Participants varied in how they understood and engaged with reproducibility. Notably, several accounts framed reproducibility in terms of the adjacent concept of openness from the outset. Some were unfamiliar with reproducibility and open science vocabulary, while others used it interchangeably with related terms such as *replicability*, *open research* and *open access*. However, when describing their research routines or best practices in their fields, many referred to actions consistent with these concepts: 52 of 60 participants described at least some OS practices to improve the transparency and accessibility of their research, while 39 of 60 reported using more direct reproducibility measures, understood here as explicit actions taken to make their own work reproducible beyond proxy practices associated with openness more broadly. Across both categories, 55 of 60 participants referred to at least one such practice.

Table 2. Themes, Main Topics, and Explanations of These Categories.

| Theme | Main topics | Explanation |
|---|---|---|
| **Navigating the research ecosystem** | Institutional incentives, resources and policies; Journal editorial priorities; Peer review; Funder policies | How institutional, publishing and funders logics create barriers, incentives and opportunities for reproducible research. |
| **Social and cultural dynamics as drivers and barriers** | Research culture and disciplinary norms; Generational differences; Research ownership and competition; Societal expectations for openness; Collaborations (academic and industry) | Influence of shared norms, values, and relationships on openness, trust, and willingness to adopt reproducible practices. |
| **Resourcing reproducibility** | Skills and training; Technical infrastructure; Guidelines and standards; Time availability; Financial resources; Awareness of reproducibility and open science | Availability of human, technical, and financial resources, and the role of awareness and training in enabling or constraining reproducible practices. |
| **Inside the research process** | Field- or methods-specific barriers; Methods specification; Research material sharing; Ethical and legal restrictions | Procedural, methodological, and contextual factors within research workflows that determine the feasibility of reproducing research. |
| **From personal commitment to shared responsibility** | Reflective and pragmatic motivations; No perceived barriers; Reproducibility accountability | Individual motivations and experiences shaped by both personal values and external expectations that influence engagement with reproducible research, and perceived responsibility for it in the research ecosystem. |

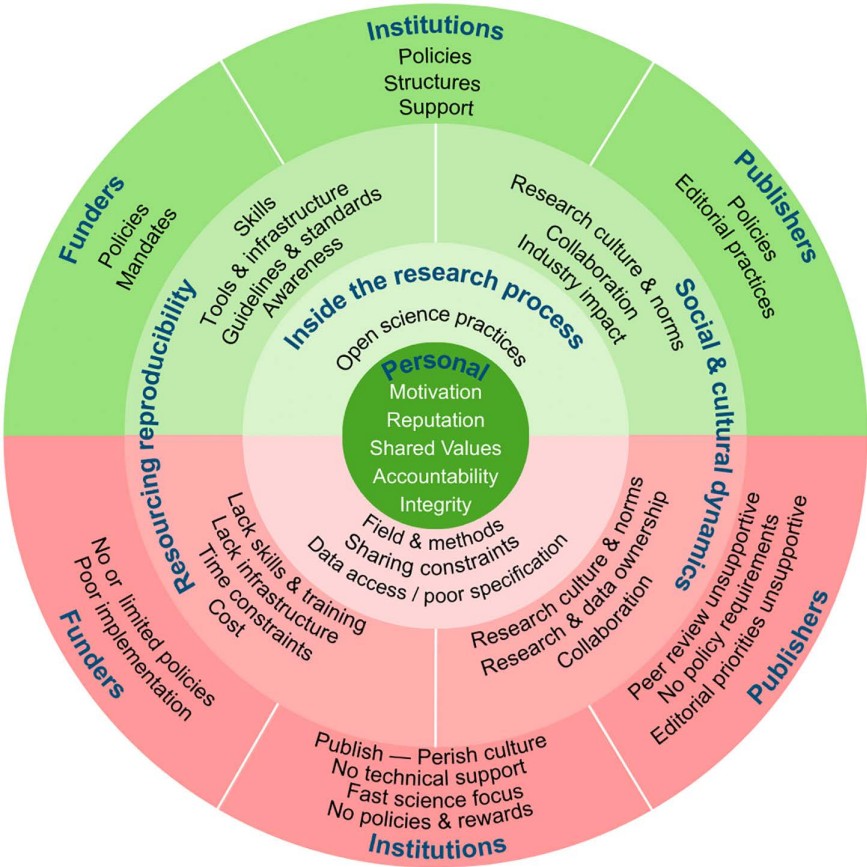

**Fig 1. Visual synthesis of barriers and facilitators across ecosystem layers.**

In the discussions, participants often moved fluidly between describing their own actions and making broader claims about disciplinary norms or approaches and behaviours of their colleagues. This suggests that researchers interpret their own practices in relation to collective expectations and field-level narratives, rather than in isolation.

## Theme 1: Navigating the research ecosystem

Participants described a complex research ecosystem where their home research institutions (hereafter referred to as institutions), journals, and funders interdependently shape the conditions for conducting research. Among them, journals and institutions were identified as the most immediate forces defining what counts, how it is rewarded, and how research is driven.

A recurring theme was the emphasis on productivity and impact in institutional settings. Participants described their immediate working environments as driven by *'publish or perish'* logic, where hiring and promotion decisions are dependent on metrics, like publication volume and journal impact factors. This focus was seen as directly incentivising fast, publishable research at the expense of more transparent and rigorous practices. Researchers reported that activities such as documenting workflows, publishing protocols, or sharing data are largely unrewarded. As one interviewee stated, *'They don't care if I publish only the result of the scientific achievement or a protocol. It doesn't make any difference for them. They just count the publications'* (AB970, natural sciences).

In this context, reproducibility was rarely dismissed as unimportant, but instead, it was often described as difficult to prioritise under existing institutional expectations. Researchers described feeling compelled to move quickly between projects and tailor their work to what is most publishable. Such institutional logic creates a reinforcement loop with the editorial priorities of top-tier journals, which were perceived to favour novel, innovative and positive findings. Pursuing replication studies or publishing negative results was described as relegated to lower-impact journals and, consequently, professionally risky. The trade-offs involved make researchers *'think twice before spending so much time on something that may not be published, or will be published in a journal that doesn't have a big impact factor'* (AB240, engineering and technology).

Journal policies themselves were a key area of focus, embodying both barriers and facilitators. On one hand, many researchers described their transparency requirements as minimal, weakly enforced, and generic. Strict editorial space limits were also cited as a significant problem, forcing authors to omit crucial methodological details. This issue was especially pronounced in relation to top-tier journals in some disciplines, which favour concise formats that are *'not ideal for our field'* (AB030, natural sciences).

On the other hand, when journals implemented clear and specific requirements, they were perceived as important facilitators. Most accounts focused on expectations around data or code sharing. Though enforcement varied, researchers generally saw the impact of these policies as positive. In natural and agricultural sciences, where material sharing is a strong norm, such policies reflect and reinforce communal values of openness, as one participant explained:

*'If someone asks you to give material, you cannot say no... then they will retract your paper. And that's very sharp. That's not something you negotiate. When you just submit a publication, you sign a material exchange document that you have to share your material with everyone else'* (AB500, agricultural sciences).

Similarly, the tendency towards more transparent and structured reporting was seen as having a positive influence. Several researchers, particularly in the medical and health fields, described domain-specific reporting standards, such as CLAIM (Checklist for Artificial Intelligence in Medical Imaging) in medical imaging, as helpful for improving transparency.

Peer review emerged as a mechanism with a profoundly dual role. When reviewers' feedback was oriented toward meeting editorial demands for brevity, it was seen as actively undermining transparency. As one researcher reported: *'the reviewer might ask them to remove them: "It's OK. It's something simple. Just remove them. We don't need them." (…) you have to kind of listen to the reviewer's comments because you want to get it accepted'* (AB300, engineering and technology). In contrast, when focused on methodological rigour, peer review was reported as a vital facilitator of quality and transparency, providing accountability and constructive feedback. In the natural sciences, peer review was valued for its corrective function:

*'(…) the reviewer looked at every piece of detail in our publication and then they commented, they made notes of all the inaccuracies or discrepancies, or problems that they thought were there. (…). I think this is very good. This is a very important point of the entire publication process'* (AB140, natural sciences).

In the social sciences, reviewers were seen as instrumental in pushing for methodological clarity and reflexivity. One researcher recalled:

*'If it's not there in your draft, a reviewer will ask, "What about the codes?", "How did you code the data?", "How did you analyse the data?", and also how your positionality impacts the way you analyse the data'* (AB720, social sciences).

Finally, some participants pointed to emerging editorial models that represent a more systematic approach to ensuring reproducibility. Though still rare, journals that have introduced active verification processes were seen as transformative. These included hiring domain experts to check protocols and re-run analyses, which *'improved [quality] in a huge way'*

(AB820, natural sciences). Another researcher described a journal partnering with a standards institute to independently check all data before publication, assessing whether '*the number is logical or whether reproducibility and uncertainty… are quoted in the right way*' (AB080, engineering and technology). These examples were presented as powerful, forward-looking solutions that embed quality control directly into the publication process.

Institutions were seen as not only crucial in creating incentive mechanisms but also in providing tangible support. The experience with university policies was viewed positively when backed up by resources, such as university repositories, open access funding, and research support. While these developments were seen as relatively new and inconsistently applied, they were emerging in both high- and lower-funded institutions, indicating broader systemic change. Participants particularly emphasised the benefit of tailored advice in navigating data documentation and repository use. As one interviewee described, getting support to '*write a ReadMe for the repository in a sort of standardised format (…) was really, really helpful*' (AB260, engineering and technology). Also, accessibility and broad expertise were perceived as beneficial, as another explained: '*drop-in sessions where we can pop in with questions... each person has their own expertise: one for open access, one for legal aspects, one for the repository*' (AB940, humanities). When resources or services were limited, fragmented, or offered without accompanying support, they were described as having little impact on research practices. As one of the interviewees explained, they have one person who is '*clearly overworked*' (AB020, social sciences), while the other reported that if they want to share the code, '*there's tools available, but there's no direct support, nobody that would come in and help me set up the sharing*' (AB100, social sciences).

Similarly, funder policies were seen as enablers when they combined expectations with resources. Researchers shared positive experiences where funders took initiatives to develop infrastructure, such as '*a platform for the availability of data*' (AB280, agricultural sciences), or allowed for funding workload related to dataset preparation and sharing. Funders' policies also contributed to norm-setting and created opportunities to build researchers' skills and awareness around open science. In particular, embedding expectations regarding '*how a project fits into open science*' (AB980, agricultural sciences) in the grant application stage was described as prompting researchers to think proactively about data availability and openness.

While public funding was seen as advancing open science, some participants noted that commercial funders could also drive reproducibility, particularly when economic stakes were high. In such contexts, funders were described as having strong incentives for '*robustness and reproducibility because the results have significant economic implications for the funders*' (AB420, medical and health sciences).

Despite these positive examples, many researchers noted that such supportive policies remain rare in local funding schemes. In several cases, funders were described as focusing primarily on legal compliance with data privacy rather than enabling open data practices. One participant emphasised that this left no practical support for sharing: '*We wouldn't necessarily even be funded to prepare a repository for it, upload it somewhere, or any of that*' (AB010, medical and health sciences). Furthermore, when funders did establish requirements for aspects such as open data, these were often reported as lacking follow-up or compliance checks. This approach was seen as limiting the policy's impact and turning it into a procedural exercise that '*feel[s] more like a tick box exercise than making a difference*' (AB840, medical and health sciences).

### Theme 2: Social and cultural dynamics as drivers and barriers

Participants frequently reflected on reproducible behaviours as embedded in social norms and interactions. The concept of the broader research culture was framed as a factor that reinforces systemic patterns. A recurring theme was, as one researcher framed it, '*that many people focus on the output of their research but less focus on reproducibility*' (AB040, natural sciences). Several participants noted that this fixation creates a troubling tolerance for problematic practices such as result-shaping, data polishing, or selective reporting that can contribute to spurious or otherwise non-reproducible results. These sometimes were encouraged through implicit signals within the community, as one of

the early career researchers described: '*So it's like either they want you to remove these data, which is not good, or to adjust it. Somebody is not telling you directly, but you see that in what they are saying, they are trying to interfere'* (AB780, natural sciences). This account also points to how authority structures in research environments can make it difficult for early career researchers to adopt reproducible practices, particularly when expectations are communicated indirectly and those in junior positions depend on senior colleagues for supervision, evaluation, and career progression [20,78,79]. Research culture was often seen as shaped by disciplinary and generational norms. Researchers in the humanities and social sciences, particularly those working with qualitative methodologies, highlighted that reproducibility and open science are either misunderstood or seen as irrelevant within their academic communities. Similar attitudes were reported outside these fields as well, influencing researchers' practices. Several participants, notably established researchers, noted that senior researchers tend to be more resistant to the emerging paradigm of openness and transparency. One reflected that '*the new generations are more open to this type of thing. (…) the older generations are more like that, that ownership of the data was more important for them maybe'* (AB280, agricultural sciences). The ownership of research and prevailing control among peers were seen as one of the barriers to material sharing. Participants frequently linked it to concerns about academic competition, the risk of being scooped, or the perceived economic value of research outputs. Such anxieties were described as shared across regional contexts: '*Every time we do an event like this, everybody's got a story about a time they gave away an idea and then they saw it in print a year later in somebody else's name'* (AB930, humanities, about international conferences).

While enduring norms still constrain change, interviewees perceived the ongoing cultural shift as a significant facilitator with two key functions. First, some highlighted how the emphasis on trust and societal impact in academic and public spaces is setting expectations about rigour and openness. Research findings were seen as no longer confined to scientific audiences, but as one interviewee said, they '*became more open (…) because of this more digital connection'* (AB380, social sciences). This external visibility was seen not as a threat but as a force that motivates researchers to be more transparent in their work. Second, researchers described a growing recognition of the value of openness within their research communities, attributed not only to generational shifts, but also to the responsible use of public funding and the practical value of data accessibility. As one participant framed it: '*Now for us, it's clear that the orientation of the development is open access (…) to provide an open access version of all the data you used or (…) the codes or methodologies you applied'* (AB660, engineering and technology).

Another social aspect that emerged in interviews was research collaborations. Although mentioned less frequently, it highlighted how social dynamics can influence research practices both in positive and problematic ways. Knowledge transfer and social learning were seen as valuable outcomes of collaborative research environments. Senior researchers described how shared projects contribute to the diffusion of responsible and reproducible research practices across institutions: '*Just by collaborating, we also learn these good practices, and then we can bring them to our institutions'* (AB380, social sciences). An early career researcher described how peer-led networks provide a space for improving research skills by '*access to information, training materials, forums where they can discuss if things are not going well, where to find information about how to fix things'* (AB960, social sciences).

The value of collaborations was also extended to joint work with industry, predominantly in engineering and technology. In this context, reproducibility was enforced by safety and economic necessity and often backed by structured standards. As one researcher explained, '*For them reproducibility is very critical (…) it is very important to prove that the results are reproducible and can be repeated'* (AB660, engineering and technology). Participants reported that this requirement promotes clearer documentation and more rigorous processes, which can be a valuable learning experience for people in academia:

'*When I moved to industry, I had to learn different standards. And it was something that became kind of a habit. And now I'm trying to implement this also in academia'.* (AB240, engineering and technology)

*'So, we might learn something from the private companies, something that we could try to use in academia maybe'.* (AB420, medical and health sciences)

Still, social dynamics in collaborations, even within one research team, can pose challenges to making research projects reproducible and transparent. Participants linked these difficulties to undocumented or inconsistent practices among team members. As one interviewee recalled, *'In the last round of revisions, we all of a sudden noticed that one thing the co-author had done was giving us problems with reproducibility'* (AB100, social sciences). Such issues could arise in inter-disciplinary collaborations, where awareness of reproducibility standards varies, or in large, fragmented projects, where, as one participant noted, *'sometimes I absolutely don't know what others do'* (AB620, engineering and technology). Finally, interviewees also mentioned that personnel turnover can disrupt continuity, leaving teams struggling to reconstruct decisions or respond to post-publication issues.

### Theme 3: Resourcing reproducibility

A recurring topic was the resource-dependent nature of practices aimed at ensuring reproducibility. Across interviews, researchers discussed the role of several resources, both psychological and physical, in shaping what is possible in day-to-day research.

Participants described skills as one of the factors that made reproducibility either slow and difficult or efficient and routine. Lack of training and inexperience were underlying many of the problems, including gaps in foundational statistical or methodological knowledge and unfamiliarity with the sharing process and infrastructure. These were often associated with a lack of formal training in OS or specific reproducibility practices, such as version control. As one researcher put it, *'I haven't received any training and I don't think my supervisees nowadays receive any training on open science. But I think they should'* (AB200, social sciences). Early, structured training, particularly at the master's or doctoral stage, was seen as embedding reproducibility into research routines rather than treating it as an add-on: *'Teach them the benefits… Give people the tools early enough so… it doesn't seem like an additional burden'* (AB100, social sciences). Among skills that were seen as crucial, participants discussed technical fluency with widely used open-source environments (Python, R) and standardised documentation formats, such as R Markdown templates. One researcher indicated that communicating research methods and data is also essential: *'(…) good writing skills… [to] explain everything around the data, metadata, and analysis'* (AB960, social sciences). However, the uneven distribution of domain-specific competence and know-how left less experienced researchers at a disadvantage: *'If you are a beginner… it's more difficult… you need some skills… experience can help'* (AB800, agricultural sciences). Some explained that they even lack knowledge on how to approach the university for specific advice. Such gaps translated into avoidance of reproducibility and openness practices perceived as too challenging to execute. Given the incentive structure traced in theme 1, these constraints unfold inside that logic rather than apart from it.

Even with well-developed skills, researchers often experienced barriers due to the technical fragility and fragmentation of infrastructure. This aspect was seen as critical in many fields, and if developed well, served as an important facilitator. A recurring concern was the sustainability and reliability of research infrastructure. For some researchers, data capture was constrained by the absence of electricity or internet in remote sites, limiting opportunities for accurate digital curation. In several fields, interviewees described missing or incompatible repositories that made it difficult to store and disseminate complex datasets, particularly large visual files. One natural scientist noted that *'although there are repositories that host such files, they often do not handle large datasets well'* (AB030, natural sciences), a limitation that often forced researchers into ad-hoc workarounds with poorer documentation. However, researchers pointed out that domain-specific repositories and digital archives are currently a promising development in many fields where visual judgment is essential. The same interviewee, for example, described how repositories for 3D models of specimens allow peers to *'rotate it however you like and see what is there'* (AB030, natural sciences). The capacity of infrastructure to handle diverse data

 

formats was another point raised by participants. Particularly in engineering contexts, researchers described problems with poor-quality data sharing systems and the absence of organised uploading platforms with built-in certification: *'(…) it's mostly you put the Excel file and then open access tickets'* (AB300, engineering and technology). Conversely, researchers working in the medical and health fields highlighted how robust infrastructures can facilitate reproducibility when they support secure, format-flexible data exchange across institutions, such as password-protected patient databases that link hospital datasets: *'There is a database in the internet where we can put our results from different type of hospitals (…) There is a secure side with our specific password and code numbers'* (AB880, medical and health sciences).

Researchers emphasised the role of various guidelines and standards in conducting and disseminating research. When these were absent or insufficiently developed, they were seen, by one participant, as *'the most vital obstacle'* (AB040, natural sciences). Even in technically advanced or data-intensive disciplines, where manuals are more widespread, several interviewees reported a lack of protocols for core processes such as *'storing the data for materials, modelling, testing and manufacturing'* (AB300, engineering and technology). Such documents were especially valued in laboratory and computational settings, where they help reduce variability and enable reproducibility across sites. As one researcher explained, *'We generally follow a protocol that enforces certain practices, and the interpretation usually offers less wiggle room'* (AB030, natural sciences). Many participants referred to internationally established standards, but also emphasised the importance of internal lab protocols or group-level procedures developed within research teams or consortia. Though often informal, these internal guidelines served dual purposes – providing clear instruction and supporting documentation for transparency and traceability in order to build on prior research.

Some interviewees highlighted that generic tools that support OS practices often lack field-specific guidelines. This creates an extra burden, even for researchers motivated to be transparent. One illustrative example came from a team using the Open Science Framework (OSF): *'OSF asked us to list every variable, but there are no guidelines in our field on how to fill that out, so it was pretty hard'* (AB400, social sciences). In this context, some researchers described the role of community-developed norms, which extended beyond individual labs or research groups. For example, one researcher gave an example of preparing terminology standardisation to ensure consistent interpretation of samples:

> *'Specifically, we took part in improving and unifying the terminology by creating the most comprehensive dictionary to date (…). So, because we have that tool, if we talk about a certain type, we all know it's the same thing and that it looks a particular way'* (AB520, natural sciences).

Time surfaced as a major yet often hidden limitation, with structural overcommitment compounded by the inherent demands of reproducibility-oriented practices. Participants emphasised that these limitations are rarely accounted for in institutional or project-based settings, while the overlapping commitments leave little time for research itself. Particularly, established researchers described being pulled in multiple directions – supervision, grant applications, teaching, and in some cases clinical duties. Even well-intentioned institutional initiatives were considered unworkable under these conditions: *'One idea could be for institutions to offer mandatory training or courses on reproducibility, but in my experience, researchers have countless other responsibilities, making it challenging to prioritise'* (AB420, medical and health sciences). These pressures were not evenly distributed as female participants raised time constraints more frequently, reflecting the gendered distribution of care responsibilities and persistent inequities in working conditions [79,80]. This pattern was visible not only in how many participants mentioned the issue (16 female versus 12 male), but also in the weight it carried in their accounts. Among those who raised it, time constraints were more prominent in interviews with women, with 30 references compared with 19 among men. Open science tasks were also experienced as disproportionately time-consuming. For qualitative researchers, preparing datasets for sharing required substantial additional labour: *'(…) it's not something that you can do during, you know, two hours… you have to prepare normally a guide, everything'* (AB380, social sciences), while for experimental and lab-based researchers, repeated procedures or equipment calibration were

often difficult within the compressed timelines. This was especially acute for externally funded project settings with tight deadlines. Researchers noted that under such conditions, they *'didn't have enough time to replicate due to the deadline of the project'* (AB840, medical and health sciences), and open science requirements were perceived as *'just one additional thing that was slowing it down'* (AB960, social sciences).

Furthermore, financial constraints were reported as direct barriers to both ensuring reproducibility and transparency, as well as reproducing research. These manifested in various ways. For instance, interviewees working with expensive equipment and large data files described rerunning analyses and experiments, or sharing data, as involving significant costs. To synthesise the range of financial challenges described, Table 3 summarises key constraint types, how they manifest in practice, and their consequences for reproducible research.

Financial constraints were raised across regions and disciplines, but were particularly emphasised by established researchers, whose roles often involve managing project budgets and long-term research agendas. Their accounts highlighted how fragmented funding models and limited support for dedicated activities exacerbate these challenges:

*'But the funding is generally piecemeal. So, it's very hard to plan in time. (…) I don't necessarily have huge resources to manage a programme of work'.* (AB010, social sciences)

*'But very few programmes give money for measurements, for measurement techniques, or for very high accuracy. So, I had a lot of problems finding funding'.* (AB080, engineering and technology)

While resource inequalities were often most pronounced in international contexts beyond Europe, many researchers experienced them firsthand through collaborations or mobility. Researchers in lower-income countries and underfunded institutions often lack subscriptions to academic databases, cannot afford open access publishing fees, and face restrictions on repeating experiments due to budget constraints. These disparities also affect access to equipment, the ability to publish in high-impact journals, and participation in international research collaborations. As a result, many researchers rely on personal networks or informal channels to access literature and conduct research, reinforcing structural exclusions and limiting the global reach and reproducibility of their work.

Awareness emerged in the resources theme as a critical factor that legitimises reproducibility as a goal, motivates investment in training, and orients researchers and the public toward what must be made transparent. Interviewees stressed that awareness is a necessary precondition for uptake: only once researchers know the importance of open and reproducible research can skills and training be meaningfully applied and available tools utilised. For example, many said that they were unaware of platforms such as Open Research Europe or OSF. Consequently, participants saw awareness

**Table 3. Financial Constraints on Reproducibility Identified in Interviews with Researchers.**

| Constraint Type | Manifestation | Resulting Consequences |
|---|---|---|
| **Cost of reproducing (internal validation or reproducibility of others' research)** | Expensive materials; lab time; software access; repeated measurements. | Researchers forgo validation efforts in their own work or reproducing others' studies. |
| **Cost of data access to reproduce studies** | Travelling to sites when access to physical specimens or datasets is needed; high paywalls of peripheral or historical databases. | Inability to verify others' results; dependence on secondary or informal sources. |
| **Cost of data sharing** | Storage fees and repository charges; formatting/conversion costs. | Researchers drop sharing data or code or use suboptimal formats. |
| **Fragmented or short-term funding, funding gaps** | Funds management for reproducibility-related tasks difficult; scarce funding for specific activities possible. | Data sharing tasks dropped at the end of projects; little planning possible. |

as depending on enabling conditions, such as visible and accessible resources, and aligned incentives, highlighting the importance of stakeholders in supporting the awareness and recognition of open and reproducible research.

**Theme 4: Inside the research process**

The theme *Inside the research process* shifts the lens from stakeholders' policies and resource provision to the internal dynamics of how research is conducted, documented, and communicated. Participants reflected on structural barriers rooted in the material, procedural, or epistemic characteristics of their research fields and described research practices that shape the feasibility of reproducing research.

Tacit know-how was a recurring example in technical and laboratory-based disciplines. Researchers stressed that the experiential knowledge required to handle particular materials, adjust experimental conditions, or make interpretive judgments during data collection cannot be fully codified in publications or protocols, as illustrated by one interviewee:

*'There are certain things that you can't even measure, and so it really is just down to experience and handling it and knowing how much to kind of pull it and release it'.* (AB440, engineering and technology)

Inherent variability in research objects or environments posed another constraint. Participants working with seasonal, biological, or climate-dependent systems explained that natural fluctuations can produce divergent results across time and sites despite strict adherence to protocols:

'*If someone has another sewage water system, to get exactly the same results, because you will have probably different rain conditions, (…) make it challenging to say like, "We did this in North of [country R] and we're going to do this in [region JR1] and it will be exactly the same"'*. (AB240, engineering and technology)

*'Plants cannot jump inside to escape. They have to cope. So, some years they produce like 10 metric tons, the next year it can be five or six'.* (AB500, agricultural sciences)

In applied technological fields, the inconsistencies in engineered artefacts also pose limitations, even when protocols are strictly followed. A researcher testing turbine blades explained that small differences in manufacturing and the physical properties of the materials used made it *'almost impossible'* to get identical results across tests or locations (AB240, engineering and technology):

*'So even when we test two blades that came from the same manufacturer, we get slightly different results. And this is mainly for how they manufacture, (…) even if it's the same material, normally it doesn't have exactly the same properties'.*

Location-bound unique data created barriers for those studying irreplaceable physical objects, such as documents or objects. This was particularly acute in the natural and social sciences and humanities, where artefacts are housed in specific museum collections and archives, making re-analysis dependent on physical access and curatorial approval:

*'One major barrier, connected to the nature of our research, is that fossils are housed in collections around the world, and these are unique specimens. We cannot simply substitute another specimen from a more convenient location'* (AB030, natural sciences)

*'There's no access, there's no digital access. And therefore, the only way is to go to the library, to consult it there in place. And we can't get it on loan. We can't get it out'.* (AB110, social sciences)

 

Researchers working with textual materials or datasets across multiple languages raised concerns about data fidelity and interpretability. One key issue reported was semantic shifts that can alter measurement validity:

*'So sometimes we just depend on using results that were produced in a different cultural background or with patients who speak different languages, and those, like those studies have not been reproduced. And I think there are some studies that might be going on, trying to reproduce or translating some treatment material and reproducing treatment effects, but then we will still have the problem with different types of outcome measurements'.* (AB050, medical and health sciences)

A social scientist conducting qualitative research questioned whether translations of data preserve meaning and nuance well enough: *'Is transcribing in foreign languages enough for transparency? Or is the data collection itself enough?'* (AB020, social sciences). In document-based research, the challenge was about accurately interpreting primary materials across languages without a reliable translation: *'So you have to look for a whole lot of very specific legislation in countries where you would not necessarily know the language. (…) even though I can Google Translate, uh, it's not very scientific. Let's say it's not very trustworthy to base your research on something that's been translated by software'.* (AB090, agricultural sciences)

Finally, researchers working in qualitative paradigms highlighted the epistemological and procedural limits of reproducibility. They argued that many components of qualitative research, such as interpretation, interview dynamics, and the researcher's own positionality, are *'more subjective… it's more about context-wise again'* (AB380, social sciences). As such, interviewees stressed that because meaning is co-constructed and grounded in interpretive judgment, the process remains contingent on who interprets the data. One social scientist reflected on this issue, saying:

*'Even if I have a high Krippendorff coefficient, I cannot guarantee somebody else would reproduce it the same way'.* (AB700, social sciences)

Beyond field-specific constraints, many participants discussed barriers and facilitators that cut across disciplines. Detailed, transparent method reporting emerged as a significant topic. Participants across the natural sciences, engineering and technology, and applied fields reported frequent omission of critical procedural information, from basic materials and sample preparation to measurement instructions, software versions, statistical justifications, or key modelling components. Such omissions often forced them into time-consuming trial-and-error reconstruction of procedures, as described by one researcher: *'In a lot of cases, it happened that I learned with a lot of testing and just experimenting how to achieve an exact level with the material. (…) It was hard'* (AB620, engineering and technology). Conversely, when method sections offered precise information, they were seen as a powerful facilitator. Participants gave concrete examples of good practice, such as providing details about samples, sharing how surveys were constructed, or describing the coding strategies used. One participant compared clear methods to step-by-step IKEA manuals:

*'(…) with better, clear, transparent communications, you can reproduce it easier. Yeah, because it's like IKEA. If you get the IKEA product in your country, or myself in my country, we can reproduce it. We can assemble it the same, right?'* (AB300, engineering and technology)

Such approaches, particularly when reinforced by journal requirements or embedded in laboratory norms, were reported as removing ambiguity, reducing reconstruction effort, and allowing others to reproduce methods across contexts without compromising fidelity. In the medical and health sciences, researchers highlighted the value of formal reporting guidelines such as those from the EQUATOR network (Enhancing the QUAlity and Transparency Of health Research). These checklists were considered a baseline for transparency and a helpful starting point for early career researchers:

*'They're very useful for a starting point, particularly for PhD students, and I suppose to demonstrate the robustness of the work. So, they're useful to show, to give a little bit of insight into the process'.* (AB010, medical and health sciences)

Research material sharing was another decisive point where the research process could either obstruct or enable reproducibility. Many participants expressed frustration with the data or study details 'available on request'. In practice, such requests often went unanswered, or replies arrived after long delays. Without reporting and formatting preparation done at the time of publication, *'you need to refresh your memory and the response may come delayed'* (AB040, natural sciences), as one researcher explained. Another recalled the realities of contacting authors: *'The one guy is over 90, and he has not communicated with me. The other two guys have been thrown out of their institute and the fourth one is on parental leave'* (AB080, engineering and technology). Another problem raised by participants was the quality of shared material. Several interviewees reported that missing parameters, poor formatting, or absent documentation frequently undermined the datasets' usability: *'It's like Excel files and then many columns of numbers. And then if you use Excel software, you can come up with some graphs, but you don't know their meaning'* (AB300, engineering and technology).

Others noted files that *'simply don't work'* (AB820, natural sciences), reflecting absent metadata or unrecorded processing steps. Against this backdrop, participants valued when raw and processed datasets were deposited in repositories, presented in standardised formats, and accompanied by complete contextual metadata. In some disciplines, such practices had become a requirement: *'No one will accept - or no good journal will definitely accept - an article without raw data. This is the norm in the field'* (AB820, natural sciences). Interviewees' accounts emphasised that usability of shared materials requires clear, practical instructions: *'telling people how to use it'* (AB010, medical and health sciences), so that others could readily interpret and apply the material. For instance, code release was widely regarded as essential, yet its utility hinged on the quality of documentation. As one researcher explained, *'Most of the time the codes are available… but you need something like a user manual, otherwise one can really get lost'* (AB680, engineering and technology). Teams that invested in well-documented libraries or toolkits, complete with explanatory notes and dependency specifications, were seen as producing outputs with far greater uptake and impact, enabling others to reproduce results without having to reconstruct workflows from scratch.

Ethical, legal, and contractual restrictions further compounded the practical challenges of research material sharing. Across qualitative, medical, and politically or economically sensitive research, participants described data privacy and confidentiality as key barriers, where contextual details risk re-identification. Some data were seen as inherently risky, such as audio files containing identifiable voices, which require technical modification before sharing: *'I think I would not like to publish the audio files…with the voice…you would have to change it technically… So yeah, I'm still thinking about whether or not I'm going to publish my audio files'* (AB050, medical and health sciences). These barriers were grounded in researchers' ethical and social responsibility towards participants and reinforced by regulations such as the General Data Protection Regulation and institutional ethics requirements. Some interviewees found pseudonymisation challenging due to unfamiliarity with guidelines, while others questioned the analytical value of heavily anonymised datasets. In medical and health sciences, certain data types, like medical scans, were reported as shareable only via secure, controlled-access servers, which are not widely available. Comparable constraints emerged in engineering and technology, where industrial or governmental collaborations often involve non-disclosure agreements prohibiting the release of *'industrial secrets'* (AB240, engineering and technology), with minimal methodological details appearing in publications and little room for negotiation. Economists similarly reported that work with licensed datasets from providers such as Bloomberg or private companies comes with strict prohibitions on sharing in any form.

### Theme 5: From personal commitment to shared responsibility

Researchers described a broad spectrum of motivations for making their work reproducible, ranging from deeply reflective commitments to science and society, to more situational responses shaped by professional norms, personal experiences,

and environmental requirements. For some, these motivations were reinforced by working in settings where they encountered no meaningful barriers to reproducibility. In describing what drives their practices, participants also reflected on where responsibility for ensuring reproducibility ultimately lies, positioning individual commitment within a shared accountability across the research ecosystem.

Communal values emerged as a common recurring driver. These perspectives framed science as a collaborative endeavour that thrives when results are openly shared among the scholarly community and research methods are transparent. For these researchers, reproducibility was part of a collective scientific ethic: *'Yesterday I sent one system for gait analysis biomechanical to another university. OK, I share with them everything. Why not? Because it's the science. It's not just for me, it's public meaning for me'* (AB360, medical and health sciences, natural sciences). This orientation was often tied to the service to society, particularly for those working in applied or clinical fields. Participants described a moral imperative to ensure the applicability of their work: *'(…) my main motivator is to be able to help the cancer patients. (…) if it's not me that develops that sensor, then, you know, let it be somebody else that can. Because ultimately, you just want it to be in the clinics'* (AB440, engineering and technology).

Many researchers also described their motivation as purely intrinsic, grounded in intellectual curiosity, integrity, and passion for good science. This was not contingent on external rewards, but on internal drivers. As one researcher described it: *'I'm never doing something because of the money, because of how to raise myself in the scientific group. (…) I believe in science. Believing in my motivation. I believe in what I do'* (AB360, medical and health sciences, natural sciences).

Another reflective strand linked openness and reproducibility to professional visibility and reputation. For some, making their work more visible by sharing it openly could *'put you a little bit more under the focus to people to see what you're doing'* (AB240, engineering and technology). For others, the emphasis was on reproducible work as a pathway to long-term credibility among colleagues and a safeguard against reputational harm: *'My great-grandfather used to say, "It's better to lose an eye than your name". (…) If I have one wrong number, people will remember the wrong number. So I will lose my name. They will not trust all the other work'* (AB080, engineering and technology). In both cases, the motivation is to be regarded as a reliable contributor to the scientific record and to maintain the integrity of the discipline.

Alongside these value-driven accounts, many researchers described that practices increasing reproducibility and openness were adopted more from habitual routines, professional conditioning, or responses to requirements, than deliberate reflection.

Efficiency was the most frequently mentioned pragmatic driver. Researchers emphasised that clear documentation and accessible materials could save substantial time for others: *'So when they can reproduce your result, they will rely on your result and if they have the code, they don't need to spend time to again make the code'* (AB680, engineering and technology). For some, this was also a competitive advantage, ensuring that others could adopt their datasets or methods quickly rather than turning to alternative sources. Efficiency gains extended to the authors themselves, reducing the need for *'a lot of hours on explaining to other people by e-mail what you've done, or discussing that with them'* (AB820, natural sciences) and also making it easier to revisit and build on their own past work.

For some, a key motivation for making their work reproducible was the desire to test the robustness of their findings and rule out error. As one participant said, *'We try to replicate the analysis using different techniques just to be aware that the results are due to the things we really found and not something randomly'* (AB860, medical and health sciences). Others explained that validity motivation comes from the need to establish shared benchmarks to ensure comparability across projects or to explore subgroup differences to better understand observed effects in clinical data.

A smaller group described their negative experiences with unusable data, undocumented code, or incomplete methods. Such frustrations often translated into more careful documentation to avoid inflicting similar difficulties on others: *'I got a code that was almost impossible to use (…). I ended up having to spend several months redoing that code. I don't want the next person to throw away all my work'* (AB240, engineering and technology).

Those early in their careers reported that fear and uncertainty motivated them to be meticulous in reporting. This concern was less about deliberate misconduct and more about avoiding the reputational appearance of sloppiness or dishonesty: *'I'm just so paranoid about misreporting something accidentally even, right? Like never mind committing form fraud on purpose and everything. I'm just like, "Oh, Christ, did I miss anything? Did I do this?"'* (AB320, natural sciences).

Finally, some researchers described reactive-passive motivations, where external requirements prompted reproducibility or OS practices. These included journal or funder mandates for sharing research materials, or internal labs' evaluation obligations. One researcher talking about the journal sharing mandate stated that, *'It would never cross my mind if they didn't ask me at the time of the submission'* (AB700, social sciences).

Across career stages and disciplines, several participants reported that they do not see or encounter any barriers to conducting reproducible research, often attributing this to favourable disciplinary cultures, institutional settings, or project designs. They described academic environments with strong norms for openness and data sharing: *'There's always a lot of incentive to publish, to make public, and to share the data. There is no gatekeeping of any sort'* (AB090, agricultural sciences). Their experiences with making research designs openly available were generally positive – *'I think publishing a method is not difficult'* (AB970, natural sciences) – with the additional time required viewed as a minor investment relative to the overall research process: *'So for data, for open science, if you spend two more days, that's OK for me'* (AB200, social sciences).

Across interviews, participants framed accountability for improving reproducibility as a shared but unevenly distributed responsibility. The majority placed primary responsibility on individual researchers. This was especially pronounced in relation to principal investigators, who were described as being directly accountable for projects' integrity. This responsibility was often seen as intrinsic to academic integrity and professional identity, and in some accounts as something that should be ingrained from early training. In this context, several interviewees highlighted senior researchers' roles in modelling good practices and mentoring junior colleagues. However, many participants stressed that researchers' ability to uphold these standards depends on structural supports and incentives.

Funders were seen as particularly powerful actors. Researchers claimed that if they implemented reproducibility measures and followed up on outputs, they could drive significant change. Others pointed out to publishers and journal editors as gatekeepers who could enforce reproducibility criteria. Policymakers and government bodies were mentioned more occasionally, linked to setting broader frameworks, evaluation criteria, and funding conditions.

While some participants saw a joint role of all stakeholders – researchers, institutions, funders, journals, and policy-makers – others emphasised the absence of a single actor with clear ultimate responsibility, leading to gaps in enforcement and follow-up. A few expressed uncertainty about who could realistically take on this role in the current decentralised research system.

## Discussion

The present research identified five interconnected themes that collectively influence reproducibility practices: navigating the research ecosystem, social and cultural dynamics, resources supporting reproducibility, research process, and individual motivations within shared frameworks. These themes highlight that reproducibility practices are not dependent on individual researcher behaviours alone but are shaped by interactions between various levels of the research ecosystem. Similarly, several scholars have argued that researchers do not operate in a vacuum but in a layered and fragmented system where multiple stakeholders exert influence [81–84]. Below, we discuss the influences that emerged at various layers, as demarcated by the identified themes.

Within the broader research ecosystem, current results align closely with previous findings and policy messages that institutional and journal incentive structures remain a dominant barrier, creating a reinforcement loop that rewards productivity and novelty over transparency and rigour [24–28,30,82,85–87]. At the same time, policies and infrastructures at the systemic level are perceived as powerful enablers when oriented at incentivising reproducible and open practices,

including peer reviewers who emerged as important players who could push for data transparency. This shows that systemic change is both possible and underway.

Within the next theme, cultural norms emerged as a subtle but persistent influencer of how researchers engage with reproducibility. Researchers interpret their own practices in relation to collective expectations: reproducibility is framed less as an individual choice than as a socially formed activity. The findings show a broader positive shift in research culture as a result of increasing emphasis on trust and societal impact in academic and public spaces. However, there still exists an implicit focus on output and productivity over reproducibility. Team collaborations, including partnering with industry, were seen as facilitating the diffusion of responsible and reproducible research practices, while at the same time carrying the potential for inconsistencies in practices within and between teams. Generational norms around ownership and protection of data, as well as the need to control data sharing in the face of academic competition, were other barriers discussed by researchers.

Beyond that, researchers perceive resources – both psychological, such as skills, and physical, like infrastructure, time, or funds – as decisive in shaping what is acknowledged and feasible in everyday research practice. Some identified problems, such as the availability of reliable data sharing systems, or defined protocols and standards, funding that accommodates various needs of reproducibility and research material sharing, or time needed for the execution of tasks related to it, contribute to the landscape of uneven support and infrastructure and have also been identified in previous research [16–23]. This highlights that some groups may struggle disproportionately to meet reproducibility standards.

Within the research process theme, characteristics inherent to different research processes – such as tacit knowledge requirements in laboratory work, inconsistencies in engineering artefacts, the effect of semantic shifts on data fidelity and interpretability in translated research work, and the interpretive nature of qualitative research – were found to create field- or method-specific challenges that generic reproducibility policies may not adequately address. Rather than viewing these as deficiencies, such differences underline the need for discipline-sensitive guidelines and supports that respect epistemic diversity while promoting transparency.

The final theme highlights the role of individual motivations in shaping engagement with reproducible practices, underscoring that systemic change cannot rely solely on external enforcement but must also nurture and leverage the internal drivers that make reproducibility meaningful and workable for researchers. Reflective motivations, grounded in values, professional identity, and responsibility to society, represent a powerful resource for reproducible research. When researchers see reproducibility as part of what it means to "do good science," they model practices that can spread through mentoring, collaboration, and community building. Pragmatic practices shaped by habits, efficiency gains, or external requirements highlight pathways for policy and infrastructure: if reproducibility aligns with efficiency, career advancement, or reputational benefits, it is more likely to be sustained in practice.

However, placing accountability and responsibility on individual researchers alone is counterintuitive. While the language of reproducibility reform is increasingly global, this study shows that how such initiatives are implemented and experienced is mediated by institutional cultures, governance, and resource availability [88,89].

## Strengths and Limitations

A key strength is the study's scale and scope: with 60 interviews across disciplines, countries, institutions and career stages, it offers one of the most diverse and comprehensive qualitative datasets on researchers' views on reproducibility currently available. Secondly, the applied qualitative approach using framework analysis enabled systematic identification of barriers and facilitators while preserving the contextual richness of participants' experiences, thus providing valuable insights that complement existing quantitative surveys and policy analyses. Further, the hybrid inductive-deductive coding approach allowed for both theory-driven and emergent insights to provide a comprehensive picture of the subject matter.

Two limitations concern our sample composition. First, while no discipline was strongly overrepresented, social sciences formed the largest subgroup. This may partly reflect differential responsiveness, as debates around reproducibility

and related OS reforms have been particularly visible in many social studies subfields, potentially making researchers in these areas more likely to engage with an interview study on the topic. However, our study aimed for disciplinary breadth rather than proportional representation.

Second, the study has a specific geographical scope. While recruitment was confined to the EU and the UK, this focus was deliberate given OSIRIS's policy orientation and still captures substantial diversity across the European research landscape. However, it also represents a geographical limit, so perspectives from the Global South and other underrepresented contexts, as well as other Western contexts, are absent. These perspectives are critical given global inequities in resources and infrastructures.

Furthermore, as we employ participant-reported data, they may not fully capture unconscious factors that shape practices. In addition, the majority of participants were not native English speakers, and this may have affected the depth and nuance of their responses despite careful attention to interview design and conduct. Finally, while our thematic analysis achieved saturation for the main themes, it cannot capture the full complexity and diversity of field-specific practices and challenges or the evolving impacts of emerging reforms.

### Implications for practice and future research

Based on the findings, there is a clear need to move beyond generic reproducibility policies toward context-sensitive approaches that account for disciplinary and methodological differences and resource constraints. This includes developing field-specific guidelines, providing adequate resources for reproducibility training and infrastructure, and fundamentally reconsidering reward and publishing systems to incentivise quality. The study suggests that successful institutional support requires dedicated resources, including skilled personnel who can provide hands-on assistance with various reproducibility and open science activities. In particular, additional support should be provided for researchers at under-resourced institutions to prevent exacerbating existing inequalities. Further, grassroots initiatives and peer-led networks and research communities should be supported as spaces where openness and transparency are modelled and rewarded so that they can become embedded in everyday practice. Communities also need to pay attention to inclusivity to ensure that emerging norms do not reinforce inequities. Cross-disciplinary and industry exchange is another promising avenue to access diverse standards and workflows that can be adapted and transferred across fields and settings.

To make these implications actionable, Table 4 summarises the key leverage points and targeted recommendations for major stakeholder groups across the research ecosystem, translating our findings into actionable entry points for policy and intervention design.

For future research, field- and setting-specific detailed examinations are needed to develop tailored solutions, explore less represented disciplinary contexts, and assess whether current approaches inadvertently exacerbate existing inequalities. Within the scope of the OSIRIS project, further analyses from this interview dataset will examine how researchers' perspectives vary across institutional, geographical, and other subgroup contexts. Additionally, longitudinal and mixed-methods research would help capture how current trends evolve with new policies and interventions.

### Conclusions

Reproducibility reform is not dependent on individual compliance alone, but on collective responsibility across the research ecosystem. At the macro level, institutions, funders, and publishers create incentive structures that can both enable and constrain transparency. At the meso level, disciplinary cultures and local institutional environments mediate how these structures are enacted in practice. At the micro level, researchers' motivations and identities shape how they navigate these pressures in their own work. In this context, researchers act not as mere executors of science policy but as agents who make strategic decisions in response to systemic incentives and barriers [90]. Understanding how they interpret and respond to these dynamics is essential for designing effective, context-sensitive policy interventions.

**Table 4. Stakeholder-specific Leverage Points and Recommendations for Institutions, Funders, Journals, Communities and Researchers.**

| Stakeholder group | Key leverage point from findings | Targeted recommendations |
|---|---|---|
| **Universities and research institutions (leadership)** | Assessment, resourcing, and coordination structures determine whether reproducibility work is recognised, feasible, and sustained. | 1. **Align promotion and assessment** criteria to recognise the labour in transparency and reproducibility workflows (e.g., reward data/code curation and sharing, reporting, replication, and reproducibility studies).<br>2. **Incorporate reproducibility-related activities** into formal workload and role expectations, so they are supported as part of normal research practice.<br>3. **Invest in sustained infrastructure with accompanying support** (workshops, consultation, maintenance), rather than fragmented tool provision. **Embed early structured** training on the relevant part of OS/reproducibility into Master's/PhD programmes, with practical workflow skills and formal recognition.<br>4. **Support cross-institutional and cross-sector collaborations** (incl. industry) with shared standards, clear governance (ownership/access), and resourced coordination to enable mutual learning. |
| **Universities and research institutions (research support units)** | Low-friction services, field-tailored guidance, and visible support pathways turn requirements into usable everyday workflows. | 1. **Provide hands-on, case-based support** by offering consultations and troubleshooting (drop-ins, project clinics), and, where possible, embed support staff in projects during key stages (planning, data management, sharing, close-out).<br>2. **Offer field-tailored guidance and reusable templates** (discipline-appropriate templates for documentation and metadata, data/code management plans, and consent/participant-information language).<br>3. **Make support visible and easy to find** by proactively communicating services (e.g., onboarding for new staff, newsletters/info days, and clear central signposting). |
| **Funders** | Resourced, proportionate requirements are the funder's main lever for change. | 1. **Pair expectations with resources** by allowing budget and time for data preparation, documentation, and sharing (e.g., funded workload for dataset curation).<br>2. **Embed expectations on openness and data availability early** in the grant process, so researchers plan proactively rather than fitting it at the end (e.g., clear prompts in application templates).<br>3. **Apply proportionate, field-sensitive requirements** (e.g., clear exemptions; alternative compliance routes).<br>4. **Use policies as capacity-building, not only compliance,** by combining expectations with follow-up support that help researchers build skills and awareness.<br>5. **Contribute to building shared infrastructure** to create reliable platforms for data availability and long-term access, rather than relying only on individual projects to find solutions.<br>6. **Contribute to norm-setting** by framing data accessibility as part of responsible stewardship of public funds. |
| **Universities and funders** | Financial feasibility is a system-level lever: stable funding models and shared infrastructure shape whether validation, access, and sharing can occur. | 1. **Reduce fragmentation and end-of-grant drop-off.** Design funding and institutional support so reproducibility-related tasks can be planned and completed throughout the project lifecycle, including after the formal grant end date (e.g., close-out resources, bridge support, preservation pathways).<br>2. **Support reproducibility where costs are structurally high.** Provide micro-grants or dedicated budget lines for high-cost reruns, repeat measurements, lab time, and high-accuracy methods, so internal validation and reproduction studies are not systematically deprioritised.<br>3. **Enable access for verification and reuse.** Reduce paywall and access barriers by supporting subscriptions, access agreements, and travel/access costs for archives, physical specimens, or restricted datasets needed to reproduce results.<br>4. **Target additional support to under-resourced institutions** (e.g., shared services, regional support hubs, or subsidised infrastructure) to avoid widening existing inequalities. |
| **Journals and publishers** | Publication standards and compliance checks shape what becomes normative research practice. | 1. **Rebalance editorial incentives toward robustness and transparency** by explicitly valuing reproducibility studies and null findings in acceptance criteria (alongside methodological quality and transparency).<br>2. **Set clear, practice-sensitive standards** on explicit reporting and sharing requirements that are tailored to study type and field and **require "usable availability"** (what is available, where, under what conditions, and how to access it), rather than generic "available on request" language.<br>3. **Make compliance checkable and supported** by resourcing implementation through editorial roles or workflows (e.g., data editors/curators, reproducibility checks where feasible, structured checklists). |
| **Research communities and grassroots communities** | Communities are the norm-and-meaning lever. | 1. **Co-develop inclusive, field-appropriate standards and shared resources** that reflect disciplinary methods and constraints (e.g., exemplars, glossaries, reusable protocols, and training materials), so expectations are clear without imposing one-size-fits-all rules.<br>2. **Normalise learning-oriented practice and constructive critique** by creating spaces (journal clubs, communities of practice, mentoring circles) where researchers can discuss limitations, uncertainty, and failures without stigma, and support practical skill development through peer learning.<br>3. **Strengthen domain-specific sharing infrastructures and pathways** by working with repositories, institutions, and funders to develop discipline-relevant sharing standards.<br>4. **Shape narratives about trust, stewardship, and societal value** by using communication and campaigns to connect reproducibility and openness to trustworthiness and public value in ways that resonate both within academia and with wider publics. |

*(Continued)*

**Table 4.** (Continued)

| Stakeholder group | Key leverage point from findings | Targeted recommendations |
|---|---|---|
| **Senior researchers (PIs, supervisors)** | Local leadership shapes whether reproducibility becomes routinised, discussable, and resilient. | 1. **Embed reproducibility** in team routines as a gradual pathway toward shared norms and values, while foregrounding efficiency gains.<br>2. **Create psychological safety to** raise concerns by setting explicit norms that questioning is acceptable and actively protecting junior researchers' voices.<br>3. **Clarify roles and standards early,** especially in collaborations, to reduce misunderstandings and minimise disruptions from turnover.<br>4. **Foster a learning-oriented culture within and across teams** by treating reproducibility as collective improvement work: encourage peer feedback, joint reflection on constraints and trade-offs, and shared learning with collaborators (including across institutions/industry). |
| **All stages researchers (including ECRs)** | Everyday routines and choices are key micro-level levers. | 1. **Adopt scalable "good enough" routines** that fit everyday work (templates, documentation standards, versioning).<br>2. **Report methods in sufficient detail** by documenting key decisions, parameters, preprocessing steps, and analytic choices so others can follow what was done.<br>3. **Be explicit about data/code sharing** by providing clear, specific statements about what can and cannot be shared, where it is available, and how to access it.<br>4. **Share materials and data in usable form, not only "available":** deposit raw and processed datasets (where feasible) in a stable repository, using standard formats and clear file structures, include complete contextual metadata and "readme" files.<br>5. **Use collective and institutional support** to reduce individual burden: join or form communities of practice and actively use support services. |

Note: Recommendations are derived from the qualitative themes and are intended to support intervention design rather than quantify prevalence.

To conclude, successful reproducibility initiatives require the engagement of multiple stakeholders and continued dialogue within and between researchers, research communities, institutions, funders, and publishers, to develop more effective, equitable, and sustainable approaches to standardising and supporting reproducible research.

## Supporting information

**S1 Appendix. COREQ Checklist.**
(PDF)

**S2 Appendix. Recruitment Strategy.**
(PDF)

**S3 Appendix. Interview Guide.**
(PDF)

**S4 Appendix. Codebook.**
(PDF)

**S5 Appendix. Quality Strategies & Positionality Statements.**
(PDF)

## Acknowledgments

The authors would like to thank all study participants for their time and contribution.

## Author contributions

**Conceptualization:** Magdalena Kozula, Nicholas J. DeVito, Patrick Onghena, Cinzia Colombo, Leonie A. Dudda, Veerle Van den Eynden.

**Data curation:** Magdalena Kozula, Nicholas J. DeVito, Patrick Onghena, Cinzia Colombo, Paula Muñoz Teno, Veerle Van den Eynden.

**Formal analysis:** Magdalena Kozula, Nicholas J. DeVito.

**Funding acquisition:** Patrick Onghena, Veerle Van den Eynden.

**Investigation:** Magdalena Kozula, Nicholas J. DeVito, Veerle Van den Eynden.

**Methodology:** Magdalena Kozula, Nicholas J. DeVito, Patrick Onghena, Leonie A. Dudda, Veerle Van den Eynden.

**Project administration:** Magdalena Kozula, Veerle Van den Eynden.

**Supervision:** Nicholas J. DeVito, Patrick Onghena, Veerle Van den Eynden.

**Validation:** Patrick Onghena, Cinzia Colombo, Veerle Van den Eynden.

**Visualization:** Magdalena Kozula.

**Writing – original draft:** Magdalena Kozula.

**Writing – review & editing:** Magdalena Kozula, Nicholas J. DeVito, Patrick Onghena, Cinzia Colombo, Veerle Van den Eynden.

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
