## [Decision Letter · Decision Letter 0]

27 Jan 2026

PONE-D-25-53630What helps and hinders reproducible research? Researchers’ perspectives from a cross-disciplinary interview studyPLOS One

Dear Dr. Kozula,

Thank you for submitting your manuscript to PLOS ONE. After careful consideration, we feel that it has merit but does not fully meet PLOS ONE’s publication criteria as it currently stands. Therefore, we invite you to submit a revised version of the manuscript that addresses the points raised during the review process.

Please respond to each point in the attached reviews -- although you need not make all changes suggested. Please especially strongly consider revising the literature review and adding a summary figure and table.

We look forward to receiving your revised manuscript.

Kind regards,

Micah Altman

Academic Editor

PLOS One

Journal Requirements:

Additional Editor Comments:

Please pay careful attention to the attached reviews, you need not make all of the suggested changes, but please respond to each point. Strongly consider revising the literature review and adding a summary figure and table.

Reviewers' comments:

Reviewer's Responses to Questions

**Comments to the Author**

1. Is the manuscript technically sound, and do the data support the conclusions?

Reviewer #1: Yes

Reviewer #2: Yes

2. Has the statistical analysis been performed appropriately and rigorously? 

Reviewer #1: N/A

Reviewer #2: No

3. Have the authors made all data underlying the findings in their manuscript fully available?

Reviewer #1: No

Reviewer #2: No

4. Is the manuscript presented in an intelligible fashion and written in standard English?

Reviewer #1: Yes

Reviewer #2: Yes

5. Review Comments to the Author

Reviewer #1: This manuscript presents the results of an interview-based qualitative study of barriers to and facilitators of practices associated with reproducible research. The sample size (n=60) is relatively large for a qualitative study, and the sampling methods are both well designed and described in a pre-registration. The sample focuses on European and UK researchers and oversamples institutions with high levels of Horizon 2020 funding, so like any sampling strategy it is not without limitations, but it is quite a bit better than most studies of this kind.

The biggest weakness of the manuscript lies in the literature review. The data collected is sound and represents a good empirical contribution to the literature, but the meaning of that data would be more clearly evident if it were better situated in existing studies. Right now the characterization of existing research on barriers to/facilitators of reproducibility-enhancing practices/bias-reducing practices is extremely brief (two sentences long) and makes two claims: 1) existing quantitative work identifies broad patterns but is not informative in re: the diversity of practices and problems, and 2) existing qualitative research is limited in scope/scale. This really more like a summary of the strengths and weakness of quant vs qual methods generally than it is a summary of the existing literature. What is needed here is a summary of the main findings and gaps in the literature. Have particular populations been studies more than others? Or particular techniques? Do too many studies rely on convenience samples? Answering questions like these would give us a much better understanding of what existing research says.

The studies cited in the brief discussion of existing research are also a bit of a weird slice of the literature, perhaps the result of keyword searching for “reproducibility” in the literature. To get a more complete picture, the authors should look for work on barriers to/facilitators of open science practices, as well as specific practices within the general suite of tools that enhance reproducibility. For example, Gownaris et al (2022) Data Science Journal on barriers experienced by early career researchers is highly relevant, as is Pownall et al (2021) Psychology of Women Quarterly. There are several relevant interview based studies on animal research generally (Fitzpatrick et al 2018 Lab Animal) and blinding/masking in animal studies specifically (Karp et al 2022 PLOS Biology). I’m sure looking at other specific practices would reveal more literature. There is also quite a bit of survey research on attitudes towards/beliefs about open science and reproducibility, only some of which is cited here. Tenopir et al (2020) PLOS ONE is a good example of a high-quality study that collected data on barriers to data sharing and other practices. Research using the Open Scholarship Survey (a modular survey tool to assess open scholarship attitudes, perceptions, and behavior of researchers) would be another good place to start, or Perrier et al’s (2020) PLOS ONE meta-synthesis of surveys on data sharing and reuse. Getting a better literature review in place is important because it would help us see what common patterns are confirmed by this research or what is novel.

In terms of the results themselves, it would be helpful to have some information about attitudes towards open science/reproducibility in order to judge how much of a problem response bias might be. For example, are the non-responders those who have not engaged with open science practices, or don’t think that there are problems with reproducibility in their fields? Knowing that would go some way towards discerning whether these are perceived barriers or actual barriers encountered by people who have tried to engage in the practices. For the interview guide it looks like data on attitudes towards reproducibility was collected, so commenting on how the attitudes in this sample align (or not) with existing survey data might help give a sense of any possible skew in the data. It also looks like at least one question was asked about the interviewees’ practices, so maybe some data on the extent to which this sampled population was or was not already engaging in practices could be presented.

Minor points:

1. The manuscript specifies that inter-coder reliability and agreement scores were calculated but does not specify the measure or agreement thresholds used.

2. Sometimes the field of the interviewee is presented along with the study number in brackets, and sometimes not. I think it would be helpful to have the field throughout.

3. Since the sample was divided into participants from highly funded institutions and a “remainder” category, I am wondering about whether there were differences between these two groups (and if so, maybe this variable should be included in brackets as well so we can see if, for example, perception of publish-or-perish culture is more prevalent at highly funded institutions)

Reviewer #2: The paper presents a comprehensive qualitative study exploring the factors that influence research reproducibility from the perspective of researchers across various disciplines and career stages. By conducting semi-structured interviews with 60 researchers affiliated with institutions in the EU and UK, the authors identify five interrelated themes: the research ecosystem (incentives/policies), social and cultural dynamics, resourcing (skills/infrastructure), the research process (methodological transparency), and personal/shared responsibility.

Strengths

I enjoyed reading the paper. The manuscript is methodologically sound and clearly written. The paper discusses a timely and policy-relevant topic.

With 60 participants across disciplines, career stages, regions, and institutional contexts, the study achieves a significant scale for qualitative research.

The study is carefully designed and well reported, including preregistration, use of COREQ, detailed sampling procedures, reflexivity statements, and clear documentation of deviations from the protocol.

Weaknesses

* While the thematic coverage is comprehensive, the manuscript sometimes reads as descriptively dense. The relative importance or leverage points among the five themes could be made clearer, particularly for policy and intervention design.

* The presentation of results is entirely text-heavy. The absence of figures makes the complex qualitative data difficult to navigate. I recommend the inclusion of high-quality figures to summarize the primary barriers and facilitators identified, which would provide a necessary visual synthesis of the 60 interviews.

* Although the study seeks diversity, it is limited to the European Union and the United Kingdom. As the authors acknowledge, reproducibility issues often have a Western orientation, and this study does not capture perspectives from the Global South.

* The data is based on interviews, which reflect researchers' perceptions and field-level narratives rather than an objective measure of their actual practices.

* While the paper mentions that power is unevenly distributed (e.g., senior vs. junior researchers), the analysis could go deeper into how these power imbalances specifically prevent early-career researchers from speaking up about non-reproducible practices they observe.

* While the manuscript describes a rigorous framework analysis, there is insufficient discussion regarding the potential for interviewer bias and social desirability bias during the data collection phase. While reflexivity is discussed in supplementary materials, its implications for data generation and interpretation are not fully integrated into the main manuscript.

Suggestions for Improvement

* While the paper concludes that coordinated action is needed, it would be beneficial to include a summary table or section with specific, targeted recommendations for each stakeholder group.

* The paper notes that women more often mentioned time constraints related to care responsibilities. Exploring this finding in more detail could help show how open science requirements may unintentionally increase gender inequalities in academia. It is unclear whether comments about time constraints were raised by multiple female participants or by only a small number of individuals. Clarifying this would help assess how widespread this issue is within the sample.

* The manuscript does not specify whether any systematic or comparative analysis (e.g., counts or patterns across groups) was conducted across the interview transcripts. Providing more detail on how responses differed by discipline, career stage, or region would strengthen the analysis and help readers better understand potential patterns across groups.

* Briefly reflect on why some disciplines feature more prominently in the data and how this affects interpretation.

6. PLOS authors have the option to publish the peer review history of their article (what does this mean?). If published, this will include your full peer review and any attached files.

Reviewer #1: **Yes:** Nicole C. Nelson

Reviewer #2: **Yes:** Sheeba Samuel

---

## [Author Response · Author response to Decision Letter 1]

16 Mar 2026

Authors’ responses to comments on Manuscript PONE-D-25-53630

Dear Academic Editor, Dear Micah Altman,

Thank you for considering our manuscript for publication in PLOS ONE. Below, we provide numbered, point-by-point responses (underlined and italicized), to the reviewers’ helpful comments. To address two of the major points of feedback, we expanded the literature review to more clearly summarise prior findings and gaps, and we strengthened the presentation of our results by adding a figure that visually synthesises the themes and a table with targeted, actionable recommendations for key stakeholder groups.

Review Comments to the Author

Reviewer: 1

Thank you for your helpful comments. We have provided our responses below in point-to-point format.

1. This manuscript presents the results of an interview-based qualitative study of barriers to and facilitators of practices associated with reproducible research. The sample size (n=60) is relatively large for a qualitative study, and the sampling methods are both well designed and described in a pre-registration. The sample focuses on European and UK researchers and oversamples institutions with high levels of Horizon 2020 funding, so like any sampling strategy it is not without limitations, but it is quite a bit better than most studies of this kind.

The biggest weakness of the manuscript lies in the literature review. The data collected is sound and represents a good empirical contribution to the literature, but the meaning of that data would be more clearly evident if it were better situated in existing studies. Right now the characterization of existing research on barriers to/facilitators of reproducibility-enhancing practices/bias-reducing practices is extremely brief (two sentences long) and makes two claims: 1) existing quantitative work identifies broad patterns but is not informative in re: the diversity of practices and problems, and 2) existing qualitative research is limited in scope/scale. This really more like a summary of the strengths and weakness of quant vs qual methods generally than it is a summary of the existing literature. What is needed here is a summary of the main findings and gaps in the literature. Have particular populations been studies more than others? Or particular techniques? Do too many studies rely on convenience samples? Answering questions like these would give us a much better understanding of what existing research says.

Response: We appreciate this helpful suggestion. We agree that the original Introduction did not sufficiently situate our findings within the existing evidence base. In the revised manuscript, we expanded the literature review to provide a fuller synthesis of what prior research finds and where it remains uneven, integrating suggested studies alongside a broader reproducibility and open science literature. Based on the key evidence-based limitations, we now frame more explicitly what our study adds through a qualitative account of barriers and facilitators to reproducibility.

2. The studies cited in the brief discussion of existing research are also a bit of a weird slice of the literature, perhaps the result of keyword searching for “reproducibility” in the literature. To get a more complete picture, the authors should look for work on barriers to/facilitators of open science practices, as well as specific practices within the general suite of tools that enhance reproducibility. For example, Gownaris et al (2022) Data Science Journal on barriers experienced by early career researchers is highly relevant, as is Pownall et al (2021) Psychology of Women Quarterly. There are several relevant interview based studies on animal research generally (Fitzpatrick et al 2018 Lab Animal) and blinding/masking in animal studies specifically (Karp et al 2022 PLOS Biology). I’m sure looking at other specific practices would reveal more literature. There is also quite a bit of survey research on attitudes towards/beliefs about open science and reproducibility, only some of which is cited here. Tenopir et al (2020) PLOS ONE is a good example of a high-quality study that collected data on barriers to data sharing and other practices. Research using the Open Scholarship Survey (a modular survey tool to assess open scholarship attitudes, perceptions, and behavior of researchers) would be another good place to start, or Perrier et al’s (2020) PLOS ONE meta-synthesis of surveys on data sharing and reuse. Getting a better literature review in place is important because it would help us see what common patterns are confirmed by this research or what is novel.

Response: We appreciate the helpful suggestions on the literature. We have carefully reviewed all recommended studies and, as outlined in our response to comment 1, expanded and restructured the literature review to incorporate a broader evidence base on barriers to and facilitators of reproducibility-enhancing and open science practices. We integrated the recommended references into the literature review where relevant, prioritising empirical evidence as appropriate. We also considered one of the suggested references (Pownall et al., 2021) to be more suitable for the Results section to contextualise our findings. This revised framing helps clarify which patterns our findings corroborate and what is distinctive about our study’s contribution.

3. In terms of the results themselves, it would be helpful to have some information about attitudes towards open science/reproducibility in order to judge how much of a problem response bias might be. For example, are the non-responders those who have not engaged with open science practices, or don’t think that there are problems with reproducibility in their fields? Knowing that would go some way towards discerning whether these are perceived barriers or actual barriers encountered by people who have tried to engage in the practices. For the interview guide it looks like data on attitudes towards reproducibility was collected, so commenting on how the attitudes in this sample align (or not) with existing survey data might help give a sense of any possible skew in the data. It also looks like at least one question was asked about the interviewees’ practices, so maybe some data on the extent to which this sampled population was or was not already engaging in practices could be presented.

Response: We recognise that participants’ orientation towards, and experiences with, reproducibility and open science is relevant for interpreting the findings. While we did not directly measure attitudes towards these topics, we addressed this concern in the revised manuscript by adding a paragraph in the Results that provides a basic quantification of how participants described their engagement with reproducibility and open science practices.

At the same time, we do not claim that this resolves the question of non-response bias. We cannot determine whether those who did not participate were less engaged with open science practices or less concerned about reproducibility in their fields. However, the invitation framed the study broadly as examining research practices, views, and experiences regarding transparency, reliability, and reproducibility, rather than appealing only to researchers already committed to reproducibility and open science, and at least in the initial recruitment stage, participants were not invited on the basis of any known interest in or commitment to these practices. In addition, among those who explicitly declined and provided a reason, the reasons mainly concerned time constraints, as indicated in the Methods section.

Minor points

4. The manuscript specifies that inter-coder reliability and agreement scores were calculated but does not specify the measure or agreement thresholds used.

Response: Thank you for raising this. We have revised the Data analysis section to clarify how inter-coder reliability and agreement were operationalised in this study. Rather than relying exclusively on a coefficient-based measure such as Cohen’s Kappa or Krippendorff’s alpha, we used a negotiated-agreement/calibration approach informed by Campbell et al. (2013) and Hemmler et al. (2020). This was more appropriate for our analytic design because our codebook was complex, combined deductive and inductive elements, allowed multiple codes to be applied to the same meaning unit, and required interpretive sensitivity to subtle meanings in semi-structured interview data. Our coding team brought different disciplinary and professional backgrounds, which was analytically valuable for identifying interpretations that might otherwise have been missed.

5. Sometimes the field of the interviewee is presented along with the study number in brackets, and sometimes not. I think it would be helpful to have the field throughout.

Response: We have included the field of interviewees in all participant quotes in the revised version as we agree that this provides valuable context to the quotations. However, as explained below in response to comment 6, we do not aim to draw any conclusions about variability between disciplines as that was not the aim of this analysis.

6. Since the sample was divided into participants from highly funded institutions and a “remainder” category, I am wondering about whether there were differences between these two groups (and if so, maybe this variable should be included in brackets as well so we can see if, for example, perception of publish-or-perish culture is more prevalent at highly funded institutions)

Response: Thank you for this helpful suggestion. We agree that differences between participants from highly funded institutions and those from the remainder category may be important and worthy of closer examination. However, subgroup comparison was not the primary aim of this article. This interview study was developed within the broader OSIRIS project, and the interview data are being used for several analyses. In the present article, our focus is specifically on cross-cutting thematic analysis of perceived and experienced barriers, motivations, and enablers related to reproducibility across disciplines. We therefore did not structure the analysis around formal subgroup comparisons by institutional funding category (or any other specific sub-category), nor did we tag quotations by any variable outside of discipline, as you suggested above, because we feel this specifically adds valuable context to quotations in the context of our study, even if not used for comparative analysis. We recognise that this is a valuable analytical direction and have been working on further analyses from this larger interview project, which will examine subgroup patterns, including differences related to institutional context, in more detail. We have now clarified this further by adding an explanation in the Methods section about the broader research objectives within the OSIRIS project, and by noting in the Implications for Practice and Future Research section that forthcoming analyses will examine subgroup patterns, including differences related to institutional context, in more detail.

Reviewer: 2

Thank you for your helpful comments. We have provided our responses below in point-to-point format.

The paper presents a comprehensive qualitative study exploring the factors that influence research reproducibility from the perspective of researchers across various disciplines and career stages. By conducting semi-structured interviews with 60 researchers affiliated with institutions in the EU and UK, the authors identify five interrelated themes: the research ecosystem (incentives/policies), social and cultural dynamics, resourcing (skills/infrastructure), the research process (methodological transparency), and personal/shared responsibility.

Strengths

I enjoyed reading the paper. The manuscript is methodologically sound and clearly written. The paper discusses a timely and policy-relevant topic.

With 60 participants across disciplines, career stages, regions, and institutional contexts, the study achieves a significant scale for qualitative research.

The study is carefully designed and well reported, including preregistration, use of COREQ, detailed sampling procedures, reflexivity statements, and clear documentation of deviations from the protocol.

Weaknesses

1. While the thematic coverage is comprehensive, the manuscript sometimes reads as descriptively dense. The relative importance or leverage points among the five themes could be made clearer, particularly for policy and intervention design.

Response: In response, we revised the manuscript to reduce descriptive density and make the policy and intervention implications more immediately visible. Specifically, we included a stakeholder-focused table (Table 4) that translates the findings into key leverage points and targeted, actionable recommendations.

2. The presentation of results is entirely text-heavy. The absence of figures makes the complex qualitative data difficult to navigate. I recommend the inclusion of high-quality figures to summarize the primary barriers and facilitators identified, which would provide a necessary visual synthesis of the 60 interviews.

Response: Thank you for this suggestion. In response, we added now a figure that visually synthesises the findings by contrasting barriers and facilitators and situating them across ecosystem layers surrounding researchers (Figure 1). We hope that, along with Table 4, these additions improve clarity and navigability while preserving the depth of the qualitative analysis.

3. Although the study seeks diversity, it is limited to the European Union and the United Kingdom. As the authors acknowledge, reproducibility issues often have a Western orientation, and this study does not capture perspectives from the Global South.

Response: Thank you for pointing this. We recognise that this study is geographically limited and does not capture perspectives from the Global South. As stated in the manuscript in the Methods section, recruitment was limited to researchers affiliated with research-performing institutions in the European Union and the United Kingdom because the findings are intended to inform European Commission policies on research. Within that scope, we aimed to ensure diversity across the European research landscape by including participants from different countries, disciplines, institution types, and career stages. We agree that the geographic focus should be kept in mind when interpreting the findings and note this as a limitation in the global generalisability of the findings in the Discussion section.

4. The data is based on interviews, which reflect researchers' perceptions and field-level narratives rather than an objective measure of their actual practices.

Response: The study is based on researchers’ accounts and therefore examines perceived and experienced barriers and facilitators to reproducibility, rather than providing an objective measure of actual barriers and facilitators. This is consistent with the qualitative design and with the aim of the study, which was to understand how researchers make sense of reproducibility, how they describe the factors that help or hinder it, and how these issues are experienced in everyday research settings. These detailed qualitative perspectives enrich existing efforts to assess the use of reproducibility-enhancing and open science practices, and the barriers to their use, at scale.

5. While the paper mentions that power is unevenly distributed (e.g., senior vs. junior researchers), the analysis could go deeper into how these power imbalances specifically prevent early-career researchers from speaking up about non-reproducible practices they observe.

Response: Thank you for this helpful suggestion. We have strengthened the analysis in the Results section by making more explicit how authority structures in research environments can constrain early-career researchers’ ability to adopt or advocate for reproducible practices, especially when expectations are communicated indirectly and junior researchers depend on senior colleagues for supervision, evaluation, and career progression, with references to previous work signalling it. We hope this revision addresses the raised

---

## [Decision Letter · Decision Letter 1]

16 Apr 2026

What helps and hinders reproducible research? Researchers’ perspectives from a cross-disciplinary interview study

PONE-D-25-53630R1

Dear Dr. Kozula,

We’re pleased to inform you that your manuscript has been judged scientifically suitable for publication and will be formally accepted for publication once it meets all outstanding technical requirements.

Kind regards,

Micah Altman

Academic Editor

PLOS One

Additional Editor Comments (optional):

Reviewers' comments:

Reviewer's Responses to Questions

**Comments to the Author**

1. If the authors have adequately addressed your comments raised in a previous round of review and you feel that this manuscript is now acceptable for publication, you may indicate that here to bypass the “Comments to the Author” section, enter your conflict of interest statement in the “Confidential to Editor” section, and submit your "Accept" recommendation.

Reviewer #1: All comments have been addressed

Reviewer #2: All comments have been addressed

2. Is the manuscript technically sound, and do the data support the conclusions?

Reviewer #1: Yes

Reviewer #2: Yes

3. Has the statistical analysis been performed appropriately and rigorously? 

Reviewer #1: N/A

Reviewer #2: N/A

4. Have the authors made all data underlying the findings in their manuscript fully available?

Reviewer #1: Yes

Reviewer #2: Yes

5. Is the manuscript presented in an intelligible fashion and written in standard English?

Reviewer #1: Yes

Reviewer #2: Yes

6. Review Comments to the Author

Reviewer #1: The revised manuscript does a much better job of situating the findings of the study in the existing literature, and the embargo on the underlying data has been removed so that it is now publicly accessible. The manuscript is also clearer now on the fact that it is not designed for a comparative analysis (something that seems important since both I and the other reviewer were originally confused on this point), and clarifies the methodological details I raised earlier. I think the manuscript is acceptable for publication as is, but if they authors were up for a bit more refinement I think it would be useful to revise the discussion in light of the new literature review. Right now the discussion is much as it was before, and it would be helpful for readers to revisit the literature introduced earlier and discuss which findings hold from previous qualitative or survey studies and which ones are conflicting/novel (and potential explanations for conflicting results to be pursued in future studies). The literature review at present sets up the rationale for the study, but the reader is left to do the heavy lifting of integrating this new study into the literature.

Reviewer #2: The authors have successfully addressed all previous comments. My only remaining suggestion is to include supporting references for the new recommendations added to Table 4.

7. PLOS authors have the option to publish the peer review history of their article (what does this mean?). If published, this will include your full peer review and any attached files.

Reviewer #1: **Yes:** Nicole C Nelson

Reviewer #2: **Yes:** Sheeba Samuel

---

## [Editor Report · Acceptance letter]

PONE-D-25-53630R1

PLOS One

Dear Dr. Kozula,

I'm pleased to inform you that your manuscript has been deemed suitable for publication in PLOS One. Congratulations! Your manuscript is now being handed over to our production team.

Kind regards,

on behalf of

Dr. Micah Altman

Academic Editor

PLOS One